# Antibiotic prescription patterns and associated symptoms in children living with HIV at Arthur Davison Children's Hospital in Ndola, Zambia

Jonathan Gwasupika[1,2,3], Davidson H. Hamer[3,4], Victor Daka[5]*,
Ephraim Chikwanda[6], David Mwakazanga[7], Ruth L. Mfune[5], Choolwe Jacobs[2]

1 Department of Clinical Sciences, Tropical Diseases Research Centre, Ndola, Zambia, 2 Department of Epidemiology and Biostatistics, School of Public Health, University of Zambia, Lusaka, Zambia, 3 Department of Global Health, School of Public Health, Boston University, Boston, Massachusetts, United States of America, 4 Department of Medicine, Section of Infectious Diseases, Boston University Chobanian & Avedisian School of Medicine, Boston, Massachusetts, United States of America, 5 Public Health Department, School of Medicine, Copperbelt University, Ndola, Zambia, 6 Department of Biomedical Sciences, Tropical Diseases Research Centre, Ndola, Zambia, 7 Department of Public Health, Tropical Diseases Research Centre, Ndola, Zambia

☺ These authors contributed equally to this work.
* dakavictorm@gmail.com

## Abstract

### Background

Children with human immunodeficiency virus (HIV) infection are disproportionately susceptible to bacterial infections. There are a wide range of antibacterial agents available to manage HIV positive children with bacterial infections. However, administration of antibiotics in most children is empirical which could lead to antimicrobial resistance.

### Objectives

This study aimed to determine commonly prescribed antibiotics and associated symptoms in children at Arthur Davison children's hospital antiretroviral therapy clinic in Ndola, Zambia.

### Methods

This was a cross-sectional study that analysed the antibiotic prescribing patterns from routinely collected secondary data at Arthur Davison children's hospital. Children diagnosed with HIV before the age of 5, actively attending antiretroviral therapy clinic identified by SmartCare software and who had taken antiretroviral therapy for at least 6 months were eligible. Data were collected from files of children who met the eligibility criteria. STATA software version 16 SE (STATA Corp., College Station, Texas, USA) was used for analysis. A p-value less than 0.05 was considered statistically significant at a confidence interval of 95%.

**Data availability statement:** The data underlying the findings from this study are within the manuscript and its supporting information files.

**Funding:** The author(s) received no specific funding for this work.

**Competing interests:** The authors declare that no competing interests exists.

## Results

From a total of 132 children included in the study, 37.9% presented with symptoms with the most common symptoms being cough (70.0%) and diarrhoea (30.0%). A larger proportion of children (62.1%) were on arbacavir/lamivudine/dolutogravr combination of antiretroviral therapy while 8.2% were on the tenoforvir alafenamide/lamivudine/dolutobravir regimen. Children who were on abacavir/lamivudine/dolutegravir regimen presented with more symptoms (48.8%) compared to those on tenofovir alafenamide/lamivudine/dolutegravir (21.0%) and tenofovir disoproxil fumarate/lamivudine/dolutegravir (18.2%) (p = 0.006). Approximately 60.0% of children presenting with symptoms were prescribed antibiotics. Co-trimoxazole was the most commonly (38.0%) prescribed, while erythromycin (2.0%) and Cephalexin (2.0%) were the least.

## Conclusions

Respiratory and gastrointestinal symptoms were the most common presentations suggestive of a suspected infection requiring antibiotic prescription in HIV-positive children on ART. Despite co-trimoxazole being the prophylactic drug among HIV-positive children, it was the most common antibiotic among children presenting with symptoms suggestive of an infection. This calls for the prudent use of co-trimoxazole to avoid its resistance.

## Introduction

The fight against the human immunodeficiency virus (HIV) in children has changed since the availability of antiretroviral therapy (ART) [1]. The implementation of initiating ART early to all pregnant women testing positive for HIV has played a role in reducing the transmission of HIV to children [2]. On the other hand, techniques of detecting HIV infection among exposed children have improved remarkably [3]. The improved access to ART in Sub-Saharan Africa (SSA) has resulted in a decline in HIV-related morbidity and mortality in children and adolescents [4,5] and a number of children are known to have HIV viral load suppression within 6 months after ART initiation [6].

The HIV paediatric program in Zambia has recorded sustained progress with over 64% of children living with HIV accessing ART by the end of 2017 [7]. Despite the many achievements made in ART treatment for children [8], some children present with opportunistic infections mainly affecting the respiratory system [9–12]. Because of the greater risk of acquiring opportunistic infections [13]. HIV-infected and exposed children are prescribed co-trimoxazole prophylaxis due to its antibacterial, antifungal and antiparasitic properties [14]. Co-trimoxazole is a combination of trimethoprim and sulphamethoxazole [15] and its effectiveness for prophylaxis has remarkably improved the survival of HIV-infected children [16].

Conditions affecting the respiratory system are responsible for more than 50% of all HIV-related morbidity and mortality in Sub-Saharan Africa [17]. However, children are prone to respiratory tract infections mostly resulting from viral causes regardless of HIV status [18]. There has been an increase in the usage of antibiotics for the treatment of febrile illnesses, especially in HIV-infected children presenting with respiratory tract infections or gastrointestinal symptoms [19]. This manner of antibiotic use without a specific indication for antimicrobial therapy, poses a huge risk for the development of antibiotic resistance especially to commonly, cheap and easily accessible antibiotics [20]. This may further result in challenges in

the treatment of infections among paediatric patients with immunosuppressive conditions such as children infected with HIV [21]. The effectiveness of ART is mainly monitored clinically by assessing if a child presents with any symptoms while on treatment [22,23]. This study aimed at determining common presenting symptoms and the associated antibiotics prescribed among HIV-infected children on ART at Arthur Davison's Children Hospital (ADCH) ART clinic.

## Materials and methods

### Study design and site

This was a cross-sectional study among children infected with HIV seeking care at the ART outpatient clinic at Arthur Davison's Children Hospital (ADCH) from January 2011 to October 2022. ADCH is the largest tertiary and only specialised paediatric hospital in northern Zambia, and it is the referral centre for specialised HIV care in the region. A total of 2600 HIV-infected children had been seen at ADCH ART clinic by October 2022, with about 35 patients reviewed daily.

### Study population and eligibility criteria

The study population was made up of children attending ART care at ADCH from 1st January 2011 to 31st October 2022. Approximately 321 files for children attending ART clinic were screened for eligibility. Children who were diagnosed with HIV before the age of 5 and actively attending ART clinic identified by SmartCare at the time of data collection regardless of duration of ART treatment and had been on ART for at least 6 months were included in the study. Children whose files were not available at the time of data collection and those whose file was present but were transferred out of ADCH were excluded. Of the 321 files reviewed, 132 were included in the study for analysis after screening for eligibility.

### Data collection

Secondary data on the antibiotic-prescribing patterns was collected using children's files. File identification numbers were generated in SmartCare, which is an electronic health record management system [24]. A total of 321 files selected for all children attending the ART clinic at ADCH were reviewed. A file is created for every HIV-infected child being seen at the ADCH ART clinic and linked to care by SmartCare. A unique identification code is generated in smart-care which is then recorded on the file. The files were pulled from the filing cabinet based on a child's file number generated in SmartCare. The system in SmartCare has been configured to only retain file numbers of clients who are active for a particular period. File numbers for all children who were 5 years and below at the time of HIV diagnosis were generated in SmartCare. Information on whether a child was seen at the ART clinic for any unscheduled visit, at least 6 months post ART initiation presenting with symptoms suggestive of an infection was documented in a structured tool. The 6 months for recording any symptom that may have suggested opportunistic infections was picked because at this point, the child may have been virally suppressed and the immune system may have been boosted thus able to withstand infections [25].

### Data management and analysis

Data were directly entered from the patient files into two password-protected electronically-structured data entry tools in kobo-collect. Each patient file was entered by two independent data entry persons. The resulting databases were then exported to Microsoft Excel and converted to STATA (STATA Corp., College Station, Texas, USA) software, version 16 SE. Codes

were written in STATA comparing the data values between the two databases. Differences were resolved concerning the source patient files.

We presented proportions of prescribed antibiotics for symptoms suggestive of infections in HIV-positive children seen at the ADCH ART clinic respectively using a pie chart. The sex of the patients was described using frequencies and percentages. Age was presented as a continuous variable and described using the median and interquartile range (IQR). To test for differences on whether a child presented with symptoms 6 months after ART initiation, the Fishers exact test was used. For continuous not normally distributed variables, the Mann-Whitney rank-sum test was used. STATA software, version 16 SE (STATA Corp., College Station, Texas, USA) was used for analysis. A p-value less than 0.05 was considered statistically significant at a confidence interval of 95% (95% CI).

### Ethical approval

Ethical approval to carry out the study was obtained from the Tropical Diseases Research Centre Ethics Committee (IRB registration number TDREC178/12/23). Authority to conduct the study was sought from the National Health Research Authority (NHRA). Permission to access patient files and information was obtained from the Copperbelt Provincial Health Office (CPHO) and ADCH management. Confidentiality of patient information was adhered to and data were de-identified prior to analysis. Informed consent was not obtained from any individual as this was a retrospective chart review.

## Results

### Sociodemographic characteristics

From a total of 321 children who were actively attending ART care at ADCH, 132 (41.1%) met the inclusion criteria and were included in the study. There were slightly more males (51.5%) than females in the study. The minimum age of a child at HIV diagnosis was 1 month with the maximum being 60 months of age over the period 2011 to 2022. The median age at HIV diagnosis was 20 months (IQR 10 – 35 months).

Out of 132 children, 37.9% presented with symptoms suggestive of an infection (Table 1). The median duration for children sampled in this study on ART was 74 months with an interquartile range of 52.0 to 101.5 months. A larger proportion of children were noted to be on abacavir/lamivudine/dolutegravir (ABC/3TC/DTG) (62.1%) followed by tenofovir alafenamide/lamivudine/dolutegravir (TAF/3TC/DTG) (28.8%) for the ART regimen (Table 1).

Common symptoms suggestive of an infection were cough (46.1%) followed by fever (22.4%) and diarrhoea (19.7%).

Children were noted to be on a combination of two nucleoside reverse transcriptase inhibitors (NRTIs) and one integrase strand transfer inhibitor (INSTI) for their ART regimen. A combination of abacavir/lamivudine/dolutegravir was seen to be initiated in the younger age group with a median age of about 16 months than children who were initiated on tenofovir/ lamivudine/ dolutegravir (Fig 1).

There was no noted difference between female children who presented for antiretroviral therapy (ART) care with complaints 6 months after ART initiation and male children (p = 0.421) (Table 2). The mean age of children at HIV diagnosis was higher among children with no complaints (25.2 months) than those with complaints (20.9 months). It was noted that children more frequently presented with complaints when on ABC/3TC/DTG regimen (48.8%) compared to those on TAF/3TC/DTG (21.0%) or TDF/3TC/DTG regimen (18.2%) (p = 0.006) (Fig 2).

Antibiotics were prescribed for 60.0% of the patients with symptoms suggestive of an infection. Co-trimoxazole was prescribed in 38.0% of them, while Erythromycin and Cephalexin were the least prescribed at 2% each. In 40% of these children, the antibiotic prescribed was not specified (Fig 3).

**Table 1. Characteristics of children initiated on ART at ADCH, Ndola, Zambia, 1st January 2011 to 31st October 2022.**

| Variable | Category | Frequency (N) | Percentage (%) |
|---|---|---|---|
| Age at HIV diagnosis (in months) | Median (IQR) | 20 (10.8 – 35.0) | |
| Duration on ART (in months) | Median (IQR) | 74 (52 – 101.5) | |
| Weight (kg) | Mean (SD) | 10.3 (5.1) | |
| Height (cm) | Median (IQR) | 75.8 (62.8 – 91.0) | |
| Sex | | | |
| | Female | 64 | 48.5 |
| | Male | 68 | 51.5 |
| ART regimen | | | |
| | ABC/3TC/DTG | 82 | 62.1 |
| | TAF/3TC/DTG | 38 | 28.8 |
| | TDF/3TC/DTG | 12 | 9.1 |
| Complaint | | | |
| | No | 82 | 62.1 |
| | Yes | 50 | 37.9 |

HIV = Human immunodeficiency virus, IQR = interquartile range, SD = standard deviation, ABC = abacavir, 3TC = lamivudine, DTG = dolutegravir, TAF = tenofovir alafenamide, TDF = tenofovir disoproxil.

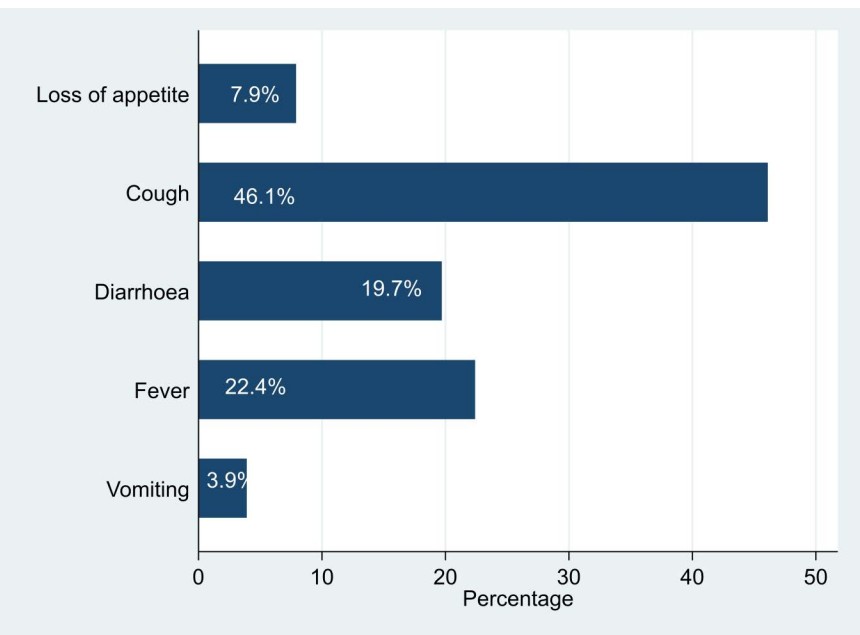

**Fig 1. Common presenting symptoms among children seeking ART services, 1st January 2011 – 31st October 2022.**

## Discussion

This study determined the common presenting symptoms and the associated antibiotics prescribed among HIV-infected children on ART at the ADCH ART clinic.

Prescribed antibiotics were specified in about 60.0% of HIV-positive children on ART, presenting with common symptoms. Various antibiotics were prescribed to treat underlying

**Table 2. Cross-tabulation of factors in children presenting with complaints, Ndola, Zambia 2011 to 2022.**

| Variable | No complaint | With complaint | p Value |
|---|---|---|---|
| Age at HIV diagnosis Mean (SD) | 20 (12 - 39) | 21 (10 - 32) | 0.330[M] |
| Duration on ART (in months) | 78 (58 – 102) | 57 (41 – 89) | 0.010[M] |
| Height (cm) Mean (SD) | 69.0 (40.2) | 63.7 (33.5) | 0.431[T] |
| Weight (Kg) Mean (SD) | 10.9 (5.6) | 9.5 (3.9) | 0.128[T] |
| Sex    Female | 42 (65.6%) | 22 (34.4%) | |
|          Male | 40 (58.8%) | 28 (41.2%) | 0.421[P] |
| Regimen    ABC/3TC/DTG | 42 (51.2%) | 40 (48.8%) | |
|             TAF/3TC/DTG | 30 (79.0%) | 8 (21.0%) | |
|             TDF/3TC/DTG | 9 (81.8%) | 2 (18.2%) | 0.006[F] |

M = Mann Whitney test, T = paired ttest, P = Pearsons Chi-square test, F = Fishers exact test.

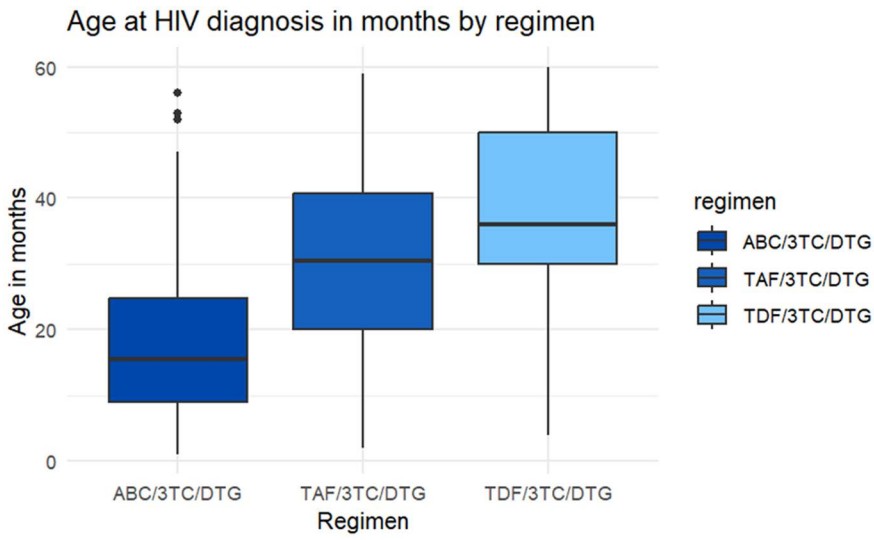

**Fig 2. Age in months at HIV diagnosis and ART regimen initiated in children at ART clinic, Ndola, Zambia, 1st January 2011- 31st October 2022.**

conditions in HIV patients. This finding was similar to a prospective cross-sectional study that showed about 65.9% of antibiotics were prescribed among HIV-positive patients in primary healthcare facilities in Mozambique [20]. Co-trimoxazole was the most common antibiotic specified to have been prescribed among children presenting with symptoms at the ART clinic. The World Health Organisation (WHO) recommends the prescription of antibiotics in the Access group of the AWaRe classification [26]. Since Cotrimoxazole belongs to the Access group of the AWaRe classification, it could have been a contributing factor to its prescription. Furthermore, Cotrimoxazole could have been a readily available and easily accessible antibiotic at the time children presented with symptoms at the ART clinic. However, Cotrimoxazole is mostly administered as a prophylactic drug against opportunistic infections among people living with HIV [27,28].

Our study found that 37.9% of the children presented with symptoms and among these, cough was the most reported symptom followed by fever and diarrhoea. Clinically, children have an insidious onset of non-productive cough and most often the temperature would be

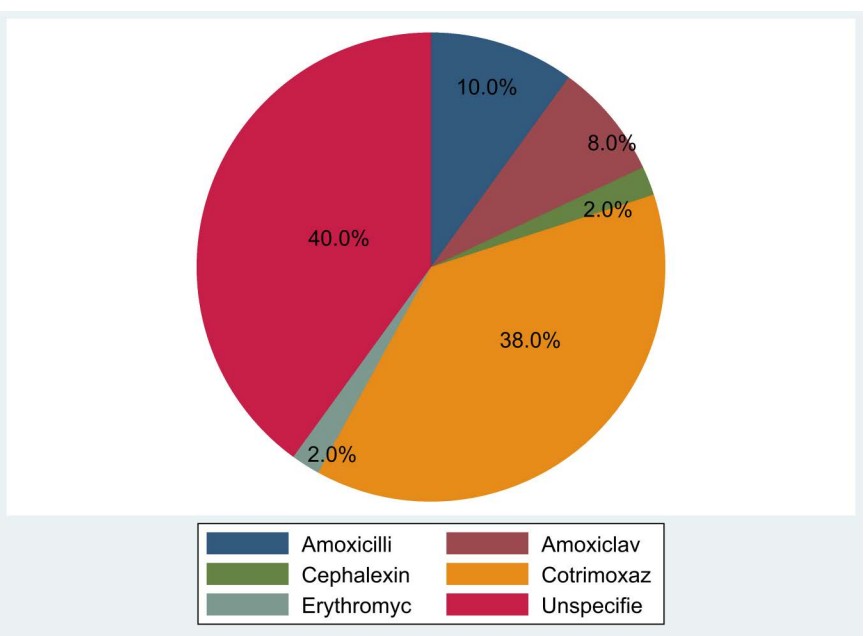

**Fig 3. Prescribed antibiotics to treat common presenting symptoms in children at ART clinic, Ndola, Zambia, 1st January 2011– 31st October 2022.**

normal for respiratory tract infection caused by viruses [29]. In this study, the majority of children presented with cough which is a symptom affecting the respiratory system. Similarly, a cohort study among adult patients admitted to the intensive care unit, it showed that severe sepsis and opportunistic infections affect organs of the respiratory system in HIV-infected patients [30]. A study in South Sudan also showed that cough (95.8%) was the most common presenting complaint among children infected with HIV [31]. An analysis of clinical characteristics of HIV-infected patients at a hospital in Malawi showed that respiratory symptoms were common with the majority of patients presenting with cough (37.5%) [32]. Coughing was most prominent since respiratory infections are the most frequently diagnosed opportunistic infections in HIV patients [33]. In children, most respiratory infections are due to viral causes with a few due to bacterial, parasitic, and fungal pathogens. About 19.7% of children seeking ART care presented with diarrhoea which was noted to be the third among the common presenting symptoms. Prior to ART, diarrhoea was reported in up to 90% of HIV-positive children residing in low-income settings and in 40 to 80% of those living in high-income countries [34]. Diarrhoeal disease has been known to affect 40% to 80% HIV HIV-infected persons and is associated with high mortality rates in sub-Saharan Africa [35]. Thus, the low proportion of patients presenting with diarrhoea found in this study could be attributed to improved access to ART and HIV care [27].

A study in Nigeria showed different findings from our study, as fever was the most common symptom (75.0%) followed by cough (65.0%) among HIV-infected children [36]. A cohort study done in India also found that fever (50.0%) was the most common presenting symptom compared to cough (43.6%) [37], which is contrary to our study findings. However, the differences can be attributed to the timing of the two studies (e.g., seasonal respiratory viral infections), sample size difference, study design and geographical location differences.

Children who were on abacavir/lamivudine/dolutegravir regimen presented with more symptoms (48.8%) compared to those on tenofovir alafenamide/lamivudine/dolutegravir

d34and tenofovir disoproxil fumarate/lamivudine/dolutobravir regimens. An important factor to treatment outcomes is adherence to ART [38]. Majority of children on abacavir/lamivudine/dolutogravir may have had poor adherence due to its twice daily administration which may have resulted in increased pill burden as compared to the other regimen in this study which are taken once.

## Limitations

This study had some limitations, one of which was the unavailability of information on the CD4 percentage and viral load to correlate with the type of regimen children were on. Secondly, analysis was done on secondary collected data thus there was minimal control on data. Additionally, missing variables could have also resulted in censoring some observations. However, the study had a good sample size to support the findings and data were well managed to ensure results are a true reflection of the population being studied. A prospective study with more variables is recommended.

## Conclusion

Respiratory symptoms were the most common presentations followed by gastrointestinal symptoms in HIV-positive children on ART, suggestive of a suspected infection requiring antibiotic prescription. Children on the abacavir/lamivudine/dolutegravir regimen presented with more symptoms than those on other regimens. Co-trimoxazole was the most commonly prescribed antibiotic for the treatment or prophylaxis of respiratory tract infections. Prescribers should be encouraged to adopt a rational use of antibiotics to reduce unnecessary prescriptions and prevent antimicrobial resistance.

## Supporting information

**S1 Data. Paeds data.**
(CSV)

## Acknowledgments

The authors would like to thank the management of Arthur Davidson Children's hospital and the Copperbelt Provincial Health Office for the permission granted to access patient information. Also acknowledge the ART clinic in charge, Ms Mercy Lukonde Malasha, for enabling easy access to the files and Ms Mercy Sulu for assisting in data collection.

## Author contributions

**Conceptualization:** Jonathan Gwasupika.

**Data curation:** Jonathan Gwasupika, Victor Daka, Ephraim Chikwanda, David Mwakazanga.

**Formal analysis:** Jonathan Gwasupika, Victor Daka, Ephraim Chikwanda, David Mwakazanga.

**Investigation:** Jonathan Gwasupika, Ephraim Chikwanda, David Mwakazanga.

**Methodology:** Jonathan Gwasupika, Davidson H. Hamer, Victor Daka, Ephraim Chikwanda, Choolwe Jacobs.

**Project administration:** Davidson H. Hamer, Choolwe Jacobs.

**Resources:** Choolwe Jacobs.

**Software:** Victor Daka, Ruth L. Mfune.

**Supervision:** Davidson H. Hamer, Choolwe Jacobs.

**Validation:** Ruth L. Mfune.

**Visualization:** David Mwakazanga.

**Writing – original draft:** Jonathan Gwasupika.

**Writing – review & editing:** Jonathan Gwasupika, Davidson H. Hamer, Victor Daka, Ephraim Chikwanda, David Mwakazanga, Ruth L. Mfune, Choolwe Jacobs.

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
