## [Decision Letter · Decision Letter 0]

12 Mar 2024

Dear Dr. Daka,

Thank you for submitting your manuscript to PLOS ONE. After careful consideration, we feel that it has merit but does not fully meet PLOS ONE’s publication criteria as it currently stands. Therefore, we invite you to submit a revised version of the manuscript that addresses the points raised during the review process.

We look forward to receiving your revised manuscript.

Kind regards,

Sarah Nanzigu, Ph.D.,MSc.,MBchB

Academic Editor

PLOS ONE

Journal Requirements:

**Additional Editor Comments:**

The manuscript needs serious revision for additional comments listed below.

I addition to comments from reviewer 2; the authors should clarify the following, and also comply to Plos one data-sharing requirements.

1. The results section of the abstract should include explicit data that include the overall proportion of children who were prescribed antibacterial agents in addition to the symptoms. Then authors can thereafter, add information related to factors that were associated with antibacterial prescription. I emphasize that is misleading to report that cotrimoxazole was the most prescribed antibacterial agent yet the study did not collect data related to prior prophylactic use of the same.

2. The conclusion of the study should rely on the study findings and there are no such findings to indicate that prescribers were irrational or careless with antibacterial use.

3.Under the methods section: The authors should throw light on what they mean with a retrospective, cross-sectional study design. They should also explain why there was one year (Nov 2022 to Oct 2023) omitted from the data collection. They collected their data during the period starting Oct 2023, but then, they only collected data ending Oct 2022. Authors need to explain why they didn't include files until 3 months prior to data collection. This anomaly needs to be explained because it creates bias in their research.

4.Still under methods; authors need to explain how they treated data for children who suffered multiple episodes of complaints during the year considered for data collection (Nov 2022 to Oct 2023). Did they collect data for the multiple episodes or for the first episode, and reasons why.

5.In order to support the selection of children who had been on ART for 3 months, authors need to include relevant literature in the background. Also prevalence data on causes of cough, fever and diarrhea should be presented for the study area. The is what authors should use this information as an indicator of the expected prevalence use of the different the antibacterial agents, and thereafter compare to their findings.

I have included the file with areas highlighted for improvement.

Reviewers' comments:

Reviewer's Responses to Questions

**Comments to the Author**

1. Is the manuscript technically sound, and do the data support the conclusions?

Reviewer #1: Yes

Reviewer #2: Partly

2. Has the statistical analysis been performed appropriately and rigorously?

Reviewer #1: Yes

Reviewer #2: No

3. Have the authors made all data underlying the findings in their manuscript fully available?

Reviewer #1: Yes

Reviewer #2: No

4. Is the manuscript presented in an intelligible fashion and written in standard English?

Reviewer #1: Yes

Reviewer #2: Yes

Reviewer #1: The authors have presented a well written manuscript with clear explanations . Some suggestions are as follows :-

1. For the title, maybe it would be better to write out the entire name of the hospital (Arthur Davison Children’s Hospital) instead of “large referral Children’s Hospital”.

2. Line no 40: - Either mention the percentage (37.9%) or the number (50/132) but not both.

3. Line no 57: - It should be “sub-Saharan Africa”.

Reviewer #2: Title: The title "Common clinical presentations associated with antibiotic use..." is not in sync with the objective which seems to focus on "common antibiotics administered and associated clinical presentations..".Throughout, there doesn't seem to be consistency in what the objective is focused on - is it the clinical presentations or the antibiotics?

Method: it is not clear if the patients took ART for 3 months or for at least 3 months - please clarify. If it's for at least 3 months, please add the median ART duration.

Results: Were there any differences in age among the three regimen groups? The conclusion that cotrimoxazole was the most prescribed drug for the clinical presentations may be misleading as it has not been established if this was given for prophylaxis or not; and if or not the patients were already on cotrimoxazole before the onset of symptoms. The results also show that children taking ABC/3TC/DTG had more complaints compared to those taking TAF or TDF/3TC/DTG. What would be important to mention is the median age of these groups of children as TAF/TDF would not be prescribed in younger children and it is known that respiratory complaints are common in young children compared to older children and adults. Such details would therefore be of interest to the readers.

Conclusion: the conclusion that "the findings suggest that caution should be applied when prescribing antibiotics" has no basis as the authors did not show any results that warrant the "caution"; although their literature review suggests development of resistance. I suggest rewording this section.

**Do you want your identity to be public for this peer review?** For information about this choice, including consent withdrawal, please see our Privacy Policy

Reviewer #1: No

Reviewer #2: No

---

## [Author Response · Author response to Decision Letter 1]

16 Apr 2024

Responses to Editor and Reviewers

1. The results section of the abstract should include explicit data that include the overall proportion of children who were prescribed antibacterial agents in addition to the symptoms. Then authors can thereafter, add information related to factors that were associated with antibacterial prescription. I emphasize that is misleading to report that cotrimoxazole was the most prescribed antibacterial agent yet the study did not collect data related to prior prophylactic use of the same.

Response: Thank you for this comment. Overall proportion of children that received antibacterials has been added in addition to symptoms. The authors note that a dose of 240mg once daily of Co-trimoxazole is given as prophylaxis and 480mg twice daily is prescribed as treatment. The Co-trimoxazole which was documented in files was prescribed as treatment.

2. The conclusion of the study should rely on the study findings and there are no such findings to indicate that prescribers were irrational or careless with antibacterial use.

Response: The comment is well noted and changes have been made.

3. Under the methods section: The authors should throw light on what they mean with a retrospective, cross-sectional study design. They should also explain why there was one year (Nov 2022 to Oct 2023) omitted from the data collection. They collected their data during the period starting Oct 2023, but then, they only collected data ending Oct 2022. Authors need to explain why they didn't include files until 3 months prior to data collection. This anomaly needs to be explained because it creates bias in their research.

Response: Thank you for the comment, the author meant review of routine collected data among HIV infected children was done as there was no prospective follow up of participants. Changes have since been made. Data were collected on children aged 5 years and below between 2011 and 2022. The earlier stated period from Nov 2022 to Oct 2023 was an error. Children were eligible regardless of duration of ART, however, data on presenting complaints were only captured if child had taken treatment for at least 6 months as this the period HIV viral load suppression is expected and not the earlier typo of 3 months and this has been updated.

4. Still under methods; authors need to explain how they treated data for children who suffered multiple episodes of complaints during the year considered for data collection (Nov 2022 to Oct 2023). Did they collect data for the multiple episodes or for the first episode, and reasons why.

Response: Analysis of data was performed on the first episode of presenting to the ART clinic with complaints despite multiple episodes of visits done by the child. The author did not consider other episodes of presentation as this required more complicated analysis of panel data

5. In order to support the selection of children who had been on ART for 3 months, authors need to include relevant literature in the background. Also prevalence data on causes of cough, fever and diarrhea should be presented for the study area. The is what authors should use this information as an indicator of the expected prevalence use of the different the antibacterial agents, and thereafter compare to their findings.

Response: This has been taken note of and relevant corrections have been made.

Reviewer #1: The authors have presented a well written manuscript with clear explanations . Some suggestions are as follows :-

1. For the title, maybe it would be better to write out the entire name of the hospital (Arthur Davison Children’s Hospital) instead of “large referral Children’s Hospital”.

Response: Changes to the title as been made as advised.

2. Line no 40: - Either mention the percentage (37.9%) or the number (50/132) but not both.

Response: This has been corrected as per suggestion

3. Line no 57: - It should be “sub-Saharan Africa”.

Response: This has been corrected as per suggestion

Reviewer #2: Title: The title "Common clinical presentations associated with antibiotic use..." is not in sync with the objective which seems to focus on "common antibiotics administered and associated clinical presentations".Throughout, there doesn't seem to be consistency in what the objective is focused on - is it the clinical presentations or the antibiotics?

Response: Thank you for the comment and authors agree. The title has been reviewed to suit the objective.

Method: it is not clear if the patients took ART for 3 months or for at least 3 months - please clarify. If it's for at least 3 months, please add the median ART duration.

Response: This has been updated to reflect the sampling approach that was employed considering the period when viral suppression takes place. Presenting complaints for Children attending ART were only entered once they were in care for at least 6 months. Therefore, children were included in the study regardless of their duration of ART.

Results: Were there any differences in age among the three regimen groups? The conclusion that cotrimoxazole was the most prescribed drug for the clinical presentations may be misleading as it has not been established if this was given for prophylaxis or not; and if or not the patients were already on cotrimoxazole before the onset of symptoms. The results also show that children taking ABC/3TC/DTG had more complaints compared to those taking TAF or TDF/3TC/DTG. What would be important to mention is the median age of these groups of children as TAF/TDF would not be prescribed in younger children and it is known that respiratory complaints are common in young children compared to older children and adults. Such details would therefore be of interest to the readers.

Response: Thank you for the comment and this was noted by the authors. Changes have been made as suggested to include median age of children on ART as well median age of children for each ART regimen.

Conclusion: the conclusion that "the findings suggest that caution should be applied when prescribing antibiotics" has no basis as the authors did not show any results that warrant the "caution"; although their literature review suggests development of resistance. I suggest rewording this section.

Response; This is well noted and conclusion has been reviewed and corrections have been made.

---

## [Decision Letter · Decision Letter 1]

3 Sep 2024

Dear Dr. Daka,

Thank you for submitting your manuscript to PLOS ONE. After careful consideration, we feel that it has merit but does not fully meet PLOS ONE’s publication criteria as it currently stands. Therefore, we invite you to submit a revised version of the manuscript that addresses the points raised during the review process.

We look forward to receiving your revised manuscript.

Kind regards,

Mabel Kamweli Aworh, DVM, MPH, PhD. FCVSN

Academic Editor

PLOS ONE

Journal Requirements:

Reviewers' comments:

Reviewer's Responses to Questions

**Comments to the Author**

Reviewer #1: All comments have been addressed

Reviewer #3: (No Response)

Reviewer #4: (No Response)

2. Is the manuscript technically sound, and do the data support the conclusions?

Reviewer #1: Yes

Reviewer #3: Yes

Reviewer #4: Partly

3. Has the statistical analysis been performed appropriately and rigorously?

Reviewer #1: Yes

Reviewer #3: Yes

Reviewer #4: Yes

4. Have the authors made all data underlying the findings in their manuscript fully available?

Reviewer #1: Yes

Reviewer #3: Yes

Reviewer #4: Yes

5. Is the manuscript presented in an intelligible fashion and written in standard English?

Reviewer #1: Yes

Reviewer #3: No

Reviewer #4: Yes

Reviewer #1: The manuscript has addressed all the comments, suggestions and modifications provided by recommenders in the previous round of reviews. HIV disproportionately affects children when it comes to bacterial infections, and the usage of antibiotics is connected to antimicrobial resistance. The authors have aimed to understand how common antibiotics are associated with an unwanted outcome, and the research was based in a children's hospital in Zambia.

Reviewer #3: Thank you for inviting me to review the manuscript titled “Association of commonly prescribed antibiotics and clinical presentation in HIV-infected children at Arthur Davison Children’s Hospital in Ndola, Zambia”. The aim of this study was to determine the most common clinical presentations and the associated antibiotics used among HIV infected children on ART at Arthur Davison’s Children Hospital ART clinic.

Below are some suggestions that the authors may find helpful:

Abstract

• Line 23: A recommendation is for the authors to consider defining HIV before using the abbreviation in the abstract and the body of the manuscript.

• Lines 29-30: Antiretroviral therapy should come before the abbreviation (ART).

• Lines 41-42: Would it be better to include the "(62.1%)" after “A larger proportion of children” on line 41, rather than the current location on line 42?

• Lines 43-45: A recommended guideline is to define abbreviations upon first appearance in the text. The definition of ABC/3TC/DTG was included in the abstract before the abbreviations were used. A recommendation is to define TDF/3TC/DTG and TAF/3TC/DTG before using their respective abbreviations.

• Lines 50-54: The abstract has a captivating conclusion.

Introduction

• Lines 75-76: The sentence appears incomplete, as a word or some words are missing. Please update as needed.

Methods

• Line 95-96: Makes mention of 2600 HIV infected children being followed at the ADCH ART clinic. It might be preferable to update the sentence to reflect the type of following that was done. A consideration could be to rephrase that portion of the sentence as “2600 HIV infected children were followed-up at ADCH ART”.

• Study population and eligibility criteria: After applying the inclusion criteria, how many participants were included in the study?

• Data collection: The manuscript previously stated that the study population for this study were HIV positive children seeking care at ADCH ART from January 2011 to October 2022. The data collection section of the manuscript however makes mention of “record files for active children attending ART from 1 October to 30 November 2022 were assessed for eligibility.” Does this imply that the final participants considered for inclusion in the study were only for children attending ART from 1 October to 30 November 2022? How many children attended the ART clinic from October 1 to November 30, 2024, and how many met the inclusion criteria? Kindly clarify information about those included in the study. What was the final number of study participants?

• Line 117: To prevent confusion on the part of the reader, it might be better to include the timeline for students attending the ART clinic as October 1st 2022 to November 30th 2022.

Data management and analysis

• The authors noted “Mann Whitney ranksum test” as a nonparametric test used in the study analyses. Kindly cross-check the preferable way for writing Man Whitney test – if it should be Mann Whitney ranksum test, Mann Whitney rank sum test or Mann Whitney rank-sum test.

Ethical approval

• Line 143: Please include the full meaning of TDRC before using the abbreviation.

Results

• Line 152: It is confusing how the number of study participants reduced from the target population of 2,600 to 321. Please include details of the selection, inclusion criteria and the final number of study participants included in the study in the methods section of the manuscript.

• Line 154: The percentage and number of males in the study as [51.5% (68/132)]. There is no need to include both the percentage and the fraction of the study population that are male. Only the percentage can be reported.

• Line 158: Same comment as for line 154. Using only the percentage should work well.

• Line 162: Consider changing “A larger proportion of children was noted” to “A larger proportion of children were noted”.

• Table 1: Some percentages do not add up to 100%. Please cross-check the figures in the manuscript for accuracy.

• Figure 1: Figure 1 is not labeled. Please label figure 1 based on the journal guidelines.

• Lines 180-182: The percentages in-text do not add up to 100%. Is some data missing or are the additional percentages not reported in-text?

• Table 2: Please cross check the P-value column of the table. Confirm the journal and citation requirements for how to write P-value. Superscripts “M”, “T”, “P” and “F” are noted on P-values in the table. A recommendation is to include a footnote that defines the abbreviations at the end of the table.

• Lines 188-189: This sentence is a bit confusing. Consider revising.

Discussion

• Lines 199-201: This sentence is confusing. Please rephrase.

• Lines 207-208: The percentage of participants with fever from the Nigerian study were included. However, the percentage of those with cough was not included. What is the percentage of HIV infected children with cough in the Nigerian study? Please include the percentage of children with cough.

• Lines 211-213: The manuscript made mention of the timing of two studies. Providing more information about the timing of the two studies and other relevant information that could be attributable to the differences noted would be helpful to the reader.

• Line 214-215: The thought in this sentence is incomplete. Some words seem to be missing.

• Line 218: A word seems to be missing on this line. Consider also adding a comma in the sentence.

• Line 221: Would it be preferable to replace “given in HIV infected people to prevent opportunistic infections” with “prescribed to in HIV infected people to prevent opportunistic infection”?

• Line 228: Replace “children born from HIV positive women” with “children born to HIV positive women”.

• Lines 237-240: The manuscript made mention of information from a Botswana paper. It is however unclear what “resistance differed between the arms in the few children studied” noted on line 240 means. Please clarify this sentence.

• Line 247: Does CD4 refer to CD4 cell count? If so, the authors can add the information for the purpose of clarity.

• Line 250: Please provide more information about what “due to lack of documentation” means in the context of the sentence on lines 248-250.

Acknowledgement

• Lines 256-258: The last five words in the first sentence seem out of place. There could be a better way to phrase the sentence “JG would like to acknowledge the Fogarty International Center and National Institute of Mental Health, of the National Institutes of Health for supporting his post-doctoral fellowship under Award Number D43 TW010543 despite not funding this study.”

Minor comments

• The purpose of this study was to determine the most common clinical presentations and the associated antibiotics used among HIV infected children on ART at Arthur Davison’s Children Hospital ART clinic. The manuscript discussed major clinical presentations. For comparison, a recommendation is to include information about the most common and the other clinical presentations of the study population.

• Cross-check the manuscript for tenses, punctuation, especially the inclusion of commas in complex sentences and spaces after full-stops.

• Tables: Cross-check the tables to confirm that correct figures are reported, and the percentages add up to 100%. Some numbers in the tables were reported with one decimal place while others were rounded up to the nearest whole number. Is there a specific reason for this? A recommendation to allow for uniformity in the percentage reported in the tables is to report the percentages in one decimal place.

Previous reviewer feedback follow-up:

3. Under the methods section: The authors should throw light on what they mean with a retrospective, cross-sectional study design. They should also explain why there was one year (Nov 2022 to Oct 2023) omitted from the data collection. They collected their data during the period starting Oct 2023, but then, they only collected data ending Oct 2022. Authors need to explain why they didn't include files until 3 months prior to data collection. This anomaly needs to be explained because it creates bias in their research.

• The authors have not fully responded to this feedback. The first part remains addressed in the manuscript.

• Retrospective or prospective are often used in describing cohort studies. This is a cross-sectional study. A recommendation is for the authors to define the study as a cross-sectional study, while providing the details of the study in the methods section to reflect why no informed consent documents were required for the study.

Conclusion: the conclusion that "the findings suggest that caution should be applied when prescribing antibiotics" has no basis as the authors did not show any results that warrant the "caution"; although their literature review suggests development of resistance. I suggest rewording this section.

• The sentence referenced in the previous feedback still features on lines 244-245. Please advise.

Thank you for the opportunity to review this manuscript. I hope the authors find the feedback helpful.

Reviewer #4: (No Response)

**Do you want your identity to be public for this peer review?** For information about this choice, including consent withdrawal, please see our Privacy Policy

Reviewer #1: No

Reviewer #3: No

Reviewer #4: **Yes: ** Dr. Oluwafolayemi Doyeni

---

## [Author Response · Author response to Decision Letter 2]

26 Oct 2024

Dear Editor.

Kindly find attached our responses to the reviers in the attached files. All authors approve the submission of the revised manuscript.

---

## [Decision Letter · Decision Letter 2]

8 Nov 2024

Dear Dr. Daka,

Thank you for submitting your manuscript to PLOS ONE. After careful consideration, we feel that it has merit but does not fully meet PLOS ONE’s publication criteria as it currently stands. Therefore, we invite you to submit a revised version of the manuscript that addresses the points raised during the review process.

We look forward to receiving your revised manuscript.

Kind regards,

Mabel Kamweli Aworh, DVM, MPH, PhD. FCVSN

Academic Editor

PLOS ONE

Journal Requirements:

Reviewers' comments:

Reviewer's Responses to Questions

**Comments to the Author**

Reviewer #3: (No Response)

Reviewer #4: All comments have been addressed

2. Is the manuscript technically sound, and do the data support the conclusions?

Reviewer #3: Partly

Reviewer #4: Yes

3. Has the statistical analysis been performed appropriately and rigorously?

Reviewer #3: Yes

Reviewer #4: Yes

4. Have the authors made all data underlying the findings in their manuscript fully available?

Reviewer #3: Yes

Reviewer #4: Yes

5. Is the manuscript presented in an intelligible fashion and written in standard English?

Reviewer #3: Yes

Reviewer #4: Yes

Reviewer #3: Thank you for the opportunity to review your manuscript on the Association of commonly prescribed antibiotics and clinical presentation in HIV infected children at Arthur Davison Children’s Hospital in Ndola, Zambia. The authors have done a good job incorporating recommendations from the previously provided feedback. Based in the updated version of the manuscript, I have some feedback and questions that I hope the authors find helpful.

Introduction

Line 58: The start of the sentence could do with some more edits. A suggestion is to replace “The implementation of initiating ART early to all pregnant women testing positive for HIV has played a role in reducing the transmission of HIV to children.” with "Early initiation of ART use among pregnant women testing positive for HIV has played a role in reducing the transmission of HIV to children.”

Lines 68-70: Consider revising the phrase “put on” with “prescribed” in the sentence below.

“Because of the greater risk of acquiring opportunistic infections, HIV-infected and exposed children are put on co-trimoxazole prophylaxis due to its antibacterial, antifungal and antiparasitic properties.”

Data collection

Line 105: Where were the children’s files from? It would be beneficial if more information is included on the source of the children’s files in the sentence ... “Secondary data on the antibiotic-prescribing patterns was collected using children’s files.”

Lines 105-114: Information from Lines 105-114 can be reorganized to ensure that the information is presented sequentially.

Line 126: How were the differences concerning the patient files resolved? Did a neutral researcher resolve the differences?

Line 151: The start of this paragraph needs more information. A recommendation for the start of this paragraph is “Out of 132 children that met the inclusion criteria, 37.9% presented with ....”

Figure 3: The colors used in the pie chart look a lot more visually appealing compared with the previous version. However, the key that provides information about the color codes in the pie chart have some alphabets missing from all the words except Cephalexin. A recommendation to correct this is to increase the length of the rectangular key box to allow for the complete words to show.

Line 97-102: It would be helpful if the authors provided more information about the students in the study. Line 102 states that “Approximately 321 files for children attending ART clinic were screened for eligibility.” This is a bit confusing because the result section referenced on Lines 146-147 identified “321 students actively attending ART care, and 132 of them meeting the inclusion criteria”. It is unclear how the potential participants went from 2,600 to 321 and finally to 132. The following questions would help them think through how to present information about how the possible sample size reduced from 2,600 to 321. Were the 2,600 children’s files screened for eligibility? How did the authors arrive at 321 potential participants from 2,600? Did 321 children’s files meet the inclusion criteria identified on lines 98-101? If yes, were a different set of inclusion criteria applied to arrive at the 132 participants included in the study? A recommendation to improve the clarity in this important section is for the authors to include a flow chart that shows the original number of “study participants”, and how the inclusion and exclusion criteria were applied to arrive at 132 participants from 2,600 participants. This will provide a foundation for the information in the data collection section.

Line 196-197: For the benefit of the reader, a recommendation is for the authors to specify the common symptoms the study participants present with.

Line 203-204: AWaRe appears for the first time in the manuscript here. Do the authors want to provide a little information about the AWaRe classification before referring to Cotrimoxazole as belonging to the Access group of the AWaRe classification? This information could be helpful for the reader to understand the full context of information presented in this paragraph.

Line 238-239: Reference was made to tenofovir alafenamide/lamivudine/dolutegravir d34, which is different from tenofovir alafenamide/lamivudine/dolutegravir used in previous sections of the manuscript. Please confirm the if the term “tenofovir alafenamide/lamivudine/dolutegravir d34" is correct, or if the authors meant to include “tenofovir alafenamide/lamivudine/dolutegravir”. Please cross-check to ensure that the correct terminologies are used to describe the respective antiretroviral therapies in the manuscript.

Minor edits: Please cross-check the manuscript to ensure that the formatting is uniform, particularly the spacing between the paragraphs and sections in the manuscript. For example, lines 237-234 and 247-252 appear to have a different line spacing format from other sections of the manuscript.

Thank you again for the opportunity to review this manuscript, I hope the authors find the feedback helpful.

Reviewer #4: (No Response)

**Do you want your identity to be public for this peer review?** For information about this choice, including consent withdrawal, please see our Privacy Policy

Reviewer #3: No

Reviewer #4: **Yes: ** Dr. Oluwafolayemi Doyeni

---

## [Author Response · Author response to Decision Letter 3]

14 Nov 2024

Thank you for the comments. The authors have provided a step by step response in the uploaded 'response to reviewers'.

---

## [Decision Letter · Decision Letter 3]

3 Dec 2024

Dear Dr. Daka,

Thank you for submitting your manuscript to PLOS ONE. After careful consideration, we feel that it has merit but does not fully meet PLOS ONE’s publication criteria as it currently stands. Therefore, we invite you to submit a revised version of the manuscript that addresses the points raised during the review process.

We look forward to receiving your revised manuscript.

Kind regards,

Mabel Kamweli Aworh, DVM, MPH, PhD. FCVSN

Academic Editor

PLOS ONE

**Journal Requirements:**

**Additional Editor Comments:**

To ensure reproducibility of a study, methods should be described with precision, transparency, and sufficient detail for another researcher to replicate the work. Please kindly provide sufficient details in the methods section.

Reviewers' comments:

Reviewer's Responses to Questions

**Comments to the Author**

Reviewer #3: (No Response)

Reviewer #4: All comments have been addressed

2. Is the manuscript technically sound, and do the data support the conclusions?

Reviewer #3: Partly

Reviewer #4: (No Response)

3. Has the statistical analysis been performed appropriately and rigorously?

Reviewer #3: Yes

Reviewer #4: (No Response)

4. Have the authors made all data underlying the findings in their manuscript fully available?

Reviewer #3: Yes

Reviewer #4: (No Response)

5. Is the manuscript presented in an intelligible fashion and written in standard English?

Reviewer #3: Yes

Reviewer #4: (No Response)

**Reviewer #3: ** Thank you again for the opportunity to review your manuscript. I have provided some feedback that I hope the authors find helpful.

On the first line in the abstract section, please confirm if immune-deficiency virus (HIV) should be immunodeficiency virus (HIV). The first sentence in the introduction section defines HIV as immunodeficiency. Please advise.

Lines 68 - 70: It could be preferable to write “Because of the greater risk of acquiring opportunistic infections, HIV-infected and exposed children are prescribed co-trimoxazole prophylaxis due to its antibacterial, antifungal and antiparasitic properties.” In place of “Because of the greater risk of acquiring opportunistic infections, HIV-infected and exposed children are prescribed on co-trimoxazole prophylaxis due to its antibacterial, antifungal and antiparasitic properties.”

Line 94: Define ADCH before using the abbreviation.

Methods: Some important information is missing in the methods section about the study population. Information provided in the methods section should be such that another researcher can reproduce the study if they follow the methods outlined in the manuscript using the same dataset used by the authors. How many children in total attended the ADCH outpatient clinic between January 11 and October 2022? I believe based on the inclusion criteria, a total of 321 files were included in the study. Information about the study population; and how based on the inclusion criteria, the final number of children included in the study was 321 should be clearly described.

Line 204: Does this phrase “Antibiotics were specified in about by 60.0% of HIV-positive children…” mean that antibiotics were used by 60.0% of HIV-positive children…? If so, please update the manuscript to prevent confusion on the part of the reader.

Limitations: What are the other limitations of using secondary data relevant to this study?

Thank you again for the opportunity to review this manuscript. I hope the authors find the feedback helpful.

**Reviewer #4:**  (No Response)

**Do you want your identity to be public for this peer review?** For information about this choice, including consent withdrawal, please see our Privacy Policy

Reviewer #3: No

Reviewer #4: **Yes: ** Dr. Oluwafolayemi Doyeni

---

## [Author Response · Author response to Decision Letter 4]

14 Dec 2024

Dear Editor,

Kindly find included in the uploaded files our response to the reviewers.

Dr Victor Daka

---

## [Editor Report · Decision Letter 4]

17 Dec 2024

Antibiotic prescription patterns and associated symptoms in children living with HIV  at Arthur Davison Children’s Hospital in Ndola, Zambia

PONE-D-24-01009R4

Dear Dr. Daka,

We’re pleased to inform you that your manuscript has been judged scientifically suitable for publication and will be formally accepted for publication once it meets all outstanding technical requirements.

Kind regards,

Mabel Kamweli Aworh, DVM, MPH, PhD. FCVSN

Academic Editor

PLOS ONE
---

## [Editor Report · Acceptance letter]

PONE-D-24-01009R4

PLOS ONE

Dear Dr. Daka,

I'm pleased to inform you that your manuscript has been deemed suitable for publication in PLOS ONE. Congratulations! Your manuscript is now being handed over to our production team.

Kind regards,

on behalf of

Dr. Mabel Kamweli Aworh

Academic Editor

PLOS ONE